# Fabrication of a Three-Dimensional Spheroid Culture System for Oral Squamous Cell Carcinomas Using a Microfabricated Device

**DOI:** 10.3390/cancers15215162

**Published:** 2023-10-26

**Authors:** Reiko Ikeda-Motonakano, Fumika Hirabayashi-Nishimuta, Naomi Yada, Ryota Yamasaki, Yoshie Nagai-Yoshioka, Michihiko Usui, Kohji Nakazawa, Daigo Yoshiga, Izumi Yoshioka, Wataru Ariyoshi

**Affiliations:** 1Division of Infections and Molecular Biology, Department of Health Promotion, Kyushu Dental University, 2-6-1 Manazuru, Kokurakita-ku, Kitakyushu, Fukuoka 803-8580, Japan; r20ikeda@fa.kyu-dent.ac.jp (R.I.-M.); r18yamasaki@fa.kyu-dent.ac.jp (R.Y.); r16yoshioka@fa.kyu-dent.ac.jp (Y.N.-Y.); 2Division of Oral Medicine, Department of Science of Physical Function, Kyushu Dental University, 2-6-1 Manazuru, Kokurakita-ku, Kitakyushu, Fukuoka 803-8580, Japan; r13hirabayashi@fa.kyu-dent.ac.jp (F.H.-N.); r11yoshiga@fa.kyu-dent.ac.jp (D.Y.); r13yoshioka@fa.kyu-dent.ac.jp (I.Y.); 3Division of Oral Pathology, Department of Health Promotion, Kyushu Dental University, 2-6-1 Manazuru, Kokurakita-ku, Kitakyushu, Fukuoka 803-8580, Japan; r12yada@fa.kyu-dent.ac.jp; 4Division of Periodontology, Department of Oral Function, Kyushu Dental University, 2-6-1 Manazuru, Kokurakita-ku, Kitakyushu, Fukuoka 803-8580, Japan; r12usui@fa.kyu-dent.ac.jp; 5Department of Life and Environment Engineering, The University of Kitakyushu, 1-1 Hibikino, Wakamatsu-ku, Kitakyushu, Fukuoka 808-0135, Japan; nakazawa@kitakyu-u.ac.jp

**Keywords:** oral squamous cell carcinoma, cancer stem cell, spheroid

## Abstract

**Simple Summary:**

Cancer stem cells (CSCs) retain their ability to self-renew and differentiate and exhibit resistance to chemotherapy and radiotherapy. Therefore, the selective eradication of CSCs is the most rational method of cancer treatment. However, the presence of CSCs in cancer tissues and cell lines is extremely low, making it difficult to isolate and collect sufficient quantities of CSCs for further studies. We used microfabrication technology to develop a device that can easily generate uniform oral cancer cell spheroids in large quantities. The spheroids produced in the microwell showed an increased expression of CSC markers and resistance to anticancer drugs, suggesting that our device could be useful for high-throughput studies on oral CSCs.

**Abstract:**

Cancer stem cells (CSCs) are considered to be responsible for recurrence, metastasis, and resistance to treatment in many types of cancers; therefore, new treatment strategies targeting CSCs are attracting attention. In this study, we fabricated a polyethylene glycol-tagged microwell device that enabled spheroid formation from human oral squamous carcinoma cells. HSC-3 and Ca9-22 cells cultured in the microwell device aggregated and generated a single spheroid per well within 24–48 h. The circular shape and smooth surface of spheroids were maintained for up to five days, and most cells comprising the spheroids were Calcein AM-positive viable cells. Interestingly, the mRNA expression of CSC markers (*Cd44*, *Oct4*, *Nanog*, and *Sox2*) were significantly higher in the spheroids than in the monolayer cultures. CSC marker-positive cells were observed throughout the spheroids. Moreover, resistance to cisplatin was enhanced in spheroid-cultured cells compared to that in the monolayer-cultured cells. Furthermore, some CSC marker genes were upregulated in HSC-3 and Ca9-22 cells that were outgrown from spheroids. In xenograft model, the tumor growth in the spheroid implantation group was comparable to that in the monolayer culture group. These results suggest that our spheroid culture system may be a high-throughput tool for producing uniform CSCs in large numbers from oral cancer cells.

## 1. Introduction

With the recent advances in treatment technologies, 5-year survival rates for breast and thyroid cancers have exceeded 90% [1]. However, the 5-year survival rate for oral cancer remains at 70% [2,3], and its high morbidity and mortality rates are problematic [4]. Therefore, establishing effective treatment strategies for oral cancer is highly necessary. Cancer stem cells (CSCs) with the capability of self-renewal contribute to tumor pathogenesis [5]. CSCs can form tumors in small numbers, are resistant to chemotherapy and radiation, and have a high probability of persisting after treatment, suggesting that they may cause tumor recurrence and metastasis [6,7]. Targeting CSCs is effective for establishing a treatment strategy for oral cancer with low recurrence and metastases. Therefore, a culture format capable of generating large numbers of CSCs is urgently needed.

Squamous cell carcinoma is the most frequent histological type of oral cancer [8,9]. CSCs in head and neck squamous cell carcinoma possess enhanced invasive and metastatic potential and resistance to therapies, contributing to lethality [10]. However, in conventional two-dimensional (2D) culture systems, the abundance of CSCs is 19.6% among all oral squamous carcinoma cells [11]. In contrast, a small percentage of CSCs are present in primary tumors, and this subpopulation possess strong CSC characteristics such as differentiation and self-renewal potential [12]. We speculated that this discrepancy was due to the deviation of the environment of 2D-cultured cells from that of in vivo cells. Three-dimensional (3D) culture methods can more accurately mimic physiological responses, genetic patterns, and other characteristics of tumors [13,14,15,16]. Among the 3D culture methods, spheroids are expected to be powerful tools for elucidating the mechanisms of tumor pathogenesis and drug discovery under conditions similar to those of in vivo tumor microenvironments [17,18]. Several methods have been developed to generate spheroids, including hanging drop cultures, culturing on non-adhesive surfaces or microfabricated microstructures, and production in rotary bioreactors [19]. However, generating uniform spheroids in large quantities is difficult, and few reports exist on spheroid cultures of oral cancer cells.

Previously, we developed a device for spheroid formation in which the bottom surface of a 500 µm diameter microwell chip was modified with polyethylene glycol (PEG) to provide a non-adhesive surface for cells [20]. The device has efficiently generated uniform spheroids in large quantities, and periodontal ligament cell spheroids generated using this device have demonstrated enhanced stemness compared to that of 2D-cultured cells [21,22]. These results led us to hypothesize that oral cancer spheroids cultured on our device would exhibit stem cell properties. Therefore, we aimed to establish a method for producing oral cancer spheroids using the fabricated microwell device and compare the characteristics of spheroid cells with those of 2D-cultured cells.

## 2. Materials and Methods

### 2.1. Cell Culture

Human tongue- (HSC-3 cells) and gingiva-derived (Ca9-22 cells) squamous carcinoma cell lineages were obtained from the Japanese Collection of Research Bioresources (JCRB; Osaka, Japan). The HSC-3 and Ca9-22 cells were maintained in Eagle’s Minimum Essential Medium (EMEM; FUJIFILM Wako Pure Chemical Co., Osaka, Japan) and high-glucose Dulbecco’s Modified Eagle Medium (DMEM; Nacalai Tesque, Kyoto, Japan), respectively, at 37 °C and in a humidified condition with 5% CO_2_. Both culture media were supplemented with 10% fetal bovine serum (FBS; Sigma-Aldrich, St. Louis, MO, USA) and 1% penicillin–streptomycin (FUJIFILM Wako Pure Chemical Co.).

### 2.2. Preparation of Microwell Device

The microwell device used for the generation of oral cancer cell spheroids was prepared as previously described [21,22]. Briefly, the surfaces of wells (195 wells) with a 500 μm diameter and 500 μm depth were coated with platinum, and the entire microwell was surrounded by a dimethylpolysiloxane (PDMS) frame. The device was treated with 2 mM PEG (MW 20,000; Tokyo Chemical Industry Co., Tokyo, Japan) to obtain nonadherent surfaces.

### 2.3. Generation of Oral Cancer Cell Spheroids

The fabricated microwell device was placed in a 35 mm dish (AGC Techno Glass Co., Shizuoka, Japan). HSC-3 and Ca9-22 cells were cultured at a concentration of 5 × 10^3^ cells/well. Briefly, a cell suspension (9.75 × 10^5^ cells/200 μL) was inoculated into the well of the device. Since 195 wells were present in the microwell device, 195 uniform spheroids consisting of 5 × 10^3^ cells were formed [21]. One hour after cell seeding, the PDMS frame was removed, and the device was tilted and incubated at 37 °C under 5% CO_2_ for up to five days. Culture medium was changed every two days. The diameter of each spheroid was measured using a microscope (BZ-X800; Keyence, Osaka, Japan). In some experiments, spheroids generated in the microwell device for three days were seeded into 35 mm dish and cultured until cells outgrown from spheroids were confluent. Culture medium was changed every two days.

### 2.4. Detection of Cell Viability in Spheroids

The viability of constituent cells in spheroids was evaluated on days 3 and 5 of culture using a Live/Dead Viability/Cytotoxicity Kit for mammalian cells (Thermo Fisher Scientific, Waltham, MA, USA) according to the manufacturer’s protocol. As the positive control for dead cells, cells were treated with 100% methanol at 37 °C for 30 min. Images of spheroids were captured using a BZ-X800 microscope and processed using BZ-II imaging software (version 1.1.2; Keyence).

### 2.5. Quantitative Real-Time Reverse Transcription Polymerase Chain Reaction (Real-Time RT-qPCR)

Total RNA was extracted from oral cancer cells cultured in monolayers or spheroids using ISOGEN II (NIPPON GENE Co., Tokyo, Japan) according to the manufacturer’s protocol. mRNA expression was analyzed via real-time RT-qPCR as previously reported [23]. The primer sequences used for real-time RT-qPCR in the present study are listed in Table 1.

### 2.6. Immunohistochemistry for CSC Markers

Oral cancer cell spheroids generated in the microwell device for three days were collected via centrifugation and suspended in 50 μL culture medium. The suspended spheroids were embedded in jellies (iPGell; GenoStaff, Tokyo, Japan) according to the manufacturer’s instructions. The jellies were immersed in a 10% formalin neutral buffer solution (FUJIFILM Wako Pure Chemical Co.) and fixed for 48 h via gentle shaking. After washing in phosphate-buffered saline (PBS) for 30 min and replacing it with 70% ethanol two times for 30 min each, the spheroids were embedded in paraffin. Paraffin blocks were sliced (4-µm thick), and sections were stained with hematoxylin (Muto pure chemicals Co., Tokyo Japan) or via immunostaining. The sections were deparaffinized with xylene and dehydrated with a series of alcohol concentrations (80%, 90%, and 100%), followed by the inactivation of endogenous peroxidase with 3% hydrogen peroxide, antigen activation with citrate buffer, and blocking with goat serum. Tissue sections were then incubated at 4 °C overnight with anti-CD44 polyclonal (1/2000; Cat. no. 15757-1-AP; Proteintech, Resemont, IL, USA), anti-Oct4 polyclonal (1/200; Cat. no. 11263-1-AP; Proteintech), anti-Nanog polyclonal (1/100; Cat. no. 14295-1-AP; Proteintech), or anti-SOX2 polyclonal (1/1000; Cat. no. 11064-1-AP; Proteintech) primary antibodies, followed by incubation with a VECTASTAIN Elite ABC kit (peroxidase, Rabbit IgG; Vector Laboratories, Newark, CA, USA) for 30 min at room temperature and staining with a Vector DAB kit (Vector Laboratories). Images of the spheroids were captured using a BZ-X800 microscope.

### 2.7. Assessment of Sensitivity to Cisplatin

Cell counting kit-8 (CCK-8; Dojindo Laboratories Co., Kumamoto, Japan) was used to evaluate the survival of the HSC-3 and Ca9-22 cells in response to cisplatin stimulation. For monolayer culture, HSC-3 and Ca9-22 cells were seeded at 2.0 × 10^4^ cells/well in a 96-well plate (AGC Techno Glass Co.) and cultured at 37 °C under 5% CO_2_ for 24 h. Subsequently, the supernatant was aspirated, and fresh medium containing cisplatin (0, 5, 20, 50, and 100 μg/mL; Tokyo Chemical Industry Co.) was added and cultured for 48 h under the same conditions. Spheroids in the microwell device on day 3 of culture were incubated for 48 h with cisplatin at the concentrations mentioned above. CCK-8 assay using WST-8 was performed according to the manufacturer’s protocol, and absorbance was measured at 450 nm using a microplate reader (Multiskan FD; Thermo Fisher Scientific). Cell viability was calculated as the percentage of viable cells in the cisplatin-free control.

### 2.8. Assessment of Tumorigenicity in Mouse Xenograft Model

Six-week-old male KSN/slc nude mice weighing 20–25 g were purchased from Japan SLC (Shizuoka, Japan). A xenograft squamous cell carcinoma model was established as described previously [24]. Briefly, monolayer- and spheroid-cultured HSC-3 cells (equivalent to 1.0 × 10^6^ cells in 0.2 mL serum-free EMEM) were subcutaneously injected into the left and right backs of the mice, respectively. Tumor size was measured daily using a digital caliper, and tumor volume was calculated using the following formula:volume (mm^3^) = long diameter (mm) × short diameter (mm) × short diameter (mm)/2.

After three weeks, the mice were sacrificed via cervical dislocation, and tumors were resected. Tumors were fixed in 10% formalin for 48 h and embedded in paraffin. The paraffin blocks were sliced in the vertical direction to the epithelium (4-μm thick sections) and stained with hematoxylin for histopathological analysis. The protocols used for the animal experiments were approved by the Kyushu Dental University Experimental Animal Care and Use Committee (permit numbers: 22-13 and 23-03).

### 2.9. Statistical Analysis

All data are expressed as mean ± standard deviation (SD). Statistical analyses were performed using Microsoft Excel. Student’s *t*-test was used to facilitate a comparison between the two groups. For comparisons among three groups, we performed a one-way analysis of variance (ANOVA) using the Tukey method. Statistical significance was set at *p* < 0.05.

## 3. Results

### 3.1. Oral Cancer Cells Aggregated and Formed Spheroids in Fabricated Microwell Device

We assessed the spheroid formation of the oral cancer cells (5000 cells/well) using the fabricated device. The HSC-3 and Ca9-22 cells cultured in our device aggregated and formed a spheroid per well within two days (HSC-3) and one day (Ca9-22), respectively. The formed spheroids persistently aggregated and maintained their spherical shape with smooth surfaces for up to five days of culture (Figure 1A,B). The mean diameter of the spheroids gradually decreased with aggregation and was approximately 150 μm on the 5th day of incubation (Figure 1C). The viability of the cancer cells in the spheroids was assessed. Calcein-AM-positive green fluorescent cells indicated live cells, while ethidium homodimer-1 (EthD-1)-positive red fluorescent cells indicated dead cells. Most of the HSC-3 and Ca9-22 cells in the spheroids were viable, and only a few dead cells were observed (Figure 1D).

### 3.2. Oral Cancer Cells in Spheroids Enhanced the Expression of CSC Markers

To compare stemness among the different culture methods, the HSC-3 and Ca9-22 cells were cultured in monolayers or a microwell device for three or five days. The mRNA expression of CSC markers, Cd44, Oct4, Nanog, and Sox2 in the spheroids was significantly higher than that in monolayer cultures on day 3 (Figure 2A). An enhanced expression of CSC marker genes, except for Cd44, in the Ca9-22 spheroid cells was maintained on day 5 of culture (Figure 2B). Immunostaining revealed the presence of CSC marker-positive cells within the spheroids. These CSC marker-positive cells were uniformly distributed in the spheroids, and no characteristic localization was observed (Figure 3).

### 3.3. Oral Cancer Cells in Spheroids Showed Increased Resistance to Anticancer Drugs

To further compare the stemness between the oral cancer cells in the 2D and 3D cultures, the HSC-3 and Ca9-22 cells cultured in the monolayer or microwell device were treated with cisplatin for 48 h. Viable cells were evaluated using CCK-8 assay. The spheroid-cultured HSC-3 cells had significantly higher percentages of viable cells after treatment with 5 and 20 μM cisplatin compared to that of the 2D-cultured cells (Figure 4A). For the Ca9-22 cells, the spheroid culture group showed resistance to high concentrations (20–100 μM) of cisplatin (Figure 4B).

### 3.4. Stemness of Oral Cancer Cells Enhanced by Spheroid Culture Was Maintained under 2D-Culture Conditions

To determine whether enhanced stemness was maintained in cells that extended and proliferated from the spheroids, the spheroids were seeded in 35 mm dishes. The oral cancer cells in the spheroids attached to the culture plate began to grow within 24 h and showed a homogeneous and spindle-shaped morphology (Figure 5A). The expression of CSC marker genes decreased in outgrown cells compared to that in the cells of the spheroids; however, the genes were highly expressed compared to those in parental 2D-cultured cells. Interestingly, Oct4 and Nanog expression in the outgrown Ca9-22 cells was markedly enhanced compared to that in the spheroids (Figure 5B).

### 3.5. The Tumorigenic Potential of Oral Cancer Spheroids Generated in the Microwell Device Was Comparable to That of Monolayer Cells

Finally, we compared the tumorigenicity of the HSC-3 cells between the 2D and 3D cultures using an in vivo xenograft model. Both the monolayer- and spheroid-cultured HSC-3 cells showed tumor-forming capabilities in the mice (Figure 6A). The tumor growth rate in the spheroid implantation group was comparable to that in the 2D culture group (Figure 6B). Furthermore, the results of our histological evaluation of tumor tissues showed no changes induced by the differences in culture methods (Figure 6C).

## 4. Discussion

For the present study, we established a culture system using a microwell device with a PEG-modified non-adhesive surface for cells to overcome the problem of size uniformity in spheroid fabrication. Oral cancer cells quickly formed uniform spherical aggregates in the wells of the device. The HSC-3 cells aggregated after 48 h, whereas the Ca9-22 cells formed spheroids within 24 h. The rate of cell aggregation may depend on the type of cancer cell line. In the present study, the number of oral cancer cells seeded per well was set at 5000 cells/well; however, periodontal ligament cells seeded in the same microwell device formed spheroids at 500 cells/well [21]. Furthermore, in studies using the hanging drop method, chondrocyte precursors [23] and glioblastoma multiforme cells [25] required a minimum of 5000 cells to aggregate, whereas islet cells [26] and renal cancer cells [27] aggregated with <500 cells. Optimizing the number of seeded cells is important for efficient spheroid fabrication using our device. The fabricated device is also capable of producing spheroids for liver cancer cells (Hep G2) [20], and it is expected to have potential applications as a tool for producing spheroids for various cancer cells in addition to oral cancer cells. Although oral cancer cell spheroids maintained their spherical morphology, their size gradually decreased as they aggregated over time. Spheroids consisting of 1600 ovarian cancer cells produced by the hanging drop method have been shown to increase in size, along with cell growth, over 17 days in culture [28]. Our device can be optimized for spheroid formation using various cells by applying microfabrication and microcontact printing technologies that allow for the setting of parameters such as well size and inter-well width. In addition to modifying the number of seeded cells, adjusting the well size of the device may induce the growth of oral cancer cell spheroids.

Oxygen supply is important for cell growth and survival. Therefore, necrosis associated with insufficient oxygen supply to cells at the center of spheroids is a significant issue. However, most oral cancer cells in spheroids generated in this study remained viable even after five days of culture, and no dead cells were observed even in the center of the spheroids. In studies with hepatocytes, central necrosis was induced within spheroids >200 μm in diameter [29]. The spheroids generated in this study were approximately 150 μm in diameter, and a certain amount of oxygen supply was probably maintained to the center of the spheroids. Furthermore, cancer tissue is prone to a hypoxic environment due to the increased oxygen consumption associated with tissue growth [30,31]. Therefore, the oral cancer cells used in this study may be able to adapt to a hypoxic microenvironment and maintain their survival and metabolism.

The expression levels of various markers are increased in CSCs. CSCs in oral squamous cell carcinomas show an increased expression of CD44, Oct4, Nanog, and Sox2, which may be utilized as candidate therapeutic targets [32,33]. The expression of CSC marker genes was higher than that in the 2D culture group, suggesting the induction of stemness in oral cancer cells that constituted spheroids. Hypoxic environment is involved in CSCs phenotypes of colon [34], renal [35], and breast [36] cancers. However, HSC-3 cells cultured under hypoxic conditions (0.5–1% O_2_ for 24 h) do not show the expression of CSC markers [37]; therefore, a hypoxic environment may not be a significant factor in the acquisition of stemness in spheroid-cultured oral cancer cells. This was supported by the immunostaining results showing that CSC marker-expressing cells were not localized in the core of spheroids, where the oxygen supply is low; however, they were uniformly distributed. CSC activation is induced by intercellular interactions and changes in the tumor microenvironment [10]. Because spheroids are 3D culture systems that do not use scaffolds or other external biomaterials [38,39], studies on the induction of stemness should focus on the microenvironment composed by the extracellular matrix, cytokines, chemokines and growth factors produced by constituent cells, and the signal responses resulting from intercellular interactions [40,41].

CSCs are characterized by an increased resistance to anticancer drugs [42]. The expression of CSC markers increases in HSC-3 cells that are resistant to cisplatin [43]. As expected, the oral cancer cells within the spheroids showed increased resistance to cisplatin. Although a decreased penetration of anticancer drugs into aggregates cannot be ruled out, the acquisition of stemness contributes to enhanced anticancer drug resistance in the constituent cells of spheroids. In addition to CSCs, various biological processes, such as autophagy, epithelial–mesenchymal transition (EMT), and metabolic reprogramming, contribute to chemotherapy resistance [44]. Our spheroid culture system will be useful for elucidating the molecular mechanisms underlying these biological processes.

To obtain cells that maintain their characteristics similar to those in tissue, cells outgrown from tissue fragments are used [45,46]. We examined whether the stemness of the spheroids was maintained in cells outgrown from the spheroids. Several CSC markers in the outgrown cells showed a lower expression than that in the spheroid cells, suggesting that the establishment of a microenvironment by spheroids contributes to the acquisition of stemness by oral cancer cells. However, the expression of some CSC markers in the outgrown cells were higher than that in the parental cells, and some markers in the outgrown cells were more upregulated than those in the spheroid cells. The kinetics of expression of CSCs markers in cells that have undergone repeated passages after outgrowth should also be validated in future studies.

A population of CSCs expressing high levels of CD44 and aldehyde dehydrogenase 1 (ALDH1) isolated from HSC-3 cells has been reported to form tumors at a higher rate (80%) than that of parental cells when 50 cells are transplanted into the oral floor of nude mice [11]. Subpopulations of oral cancer cells that highly express CD44 and show enhanced epidermal surface antigen (ESA) [47] or an elevated expression of stage-specific embryonic antigen-4 (SSEA-4) [48] have the characteristics of CSCs and show high tumorigenic potential upon transplantation (CD44^high^/ESA^high^; 5 × 10^3^ cells, CD44^high^/SSEA-4^high^; 1 × 10^4^–2 × 10^5^ cells) into the tongue of immunodeficient mice. Furthermore, the transplantation of oral cancer cell spheroids (5 × 10^5^ cells) into the buccal mucosa significantly increases tumor volume compared to that by monolayer-cultured cells [49]. However, in the xenograft model using HSC-3 cells in this study, no significant differences were observed between the monolayer and spheroid culture groups in terms of tumor size or histopathological findings. The missing characteristic of the oral cancer spheroids can be attributed to the higher number of transplanted cells (1 × 10^6^ cells) than that in previous reports and differences in transplantation sites (back vs. oral mucosa). Tumors formed by the transplantation of spheroid cells show increased resistance to cisplatin [45]. Experiments are currently underway to determine the expression of CSC markers and sensitivity to chemotherapy in tumors formed after spheroid transplantation.

The spheroids generated in this study were a monocellular system and did not have the multicellular characteristics of in vivo tumor tissues. The generation of multicellular spheroids aimed at reproducing the heterogeneous microenvironment of tumor tissue and its robust desmoplasia in vitro has been reported in various cancer cells, such as lung [50], ovarian [51], and pancreatic [52] cancers. We also found that the expression of stem cell markers and tissue regeneration potential of periodontal ligament cell spheroids are enhanced via coculturing with vascular endothelial cells [22]. Cocultured spheroids containing endothelial cells and fibroblasts may be able to reproduce the behavior of oral cancer cells in vitro in a microenvironment similar to that of living organisms. The proliferative potential and stemness of cells within spheroids change over time [53], suggesting that they are less robust than organoid cultures. Therefore, reproducing genomic and multicellular profiles of original tumor tissues is necessary by generating spheroids from primary oral cancer cells. The generation of tumor spheroids from a patient’s own cells may enable personalized approaches to screen and select appropriate drugs for patients.

## 5. Conclusions

The spheroid culture system using our fabricated microwell device can be applied as a tool to elucidate the molecular mechanisms involved in the transformation of oral cancer cells into stem cells and development of therapeutic resistance in CSCs. This device could also be useful in high-throughput analyses (e.g., in the screening of CSC-targeting drug candidates).

## Figures and Tables

**Figure 1 cancers-15-05162-f001:**
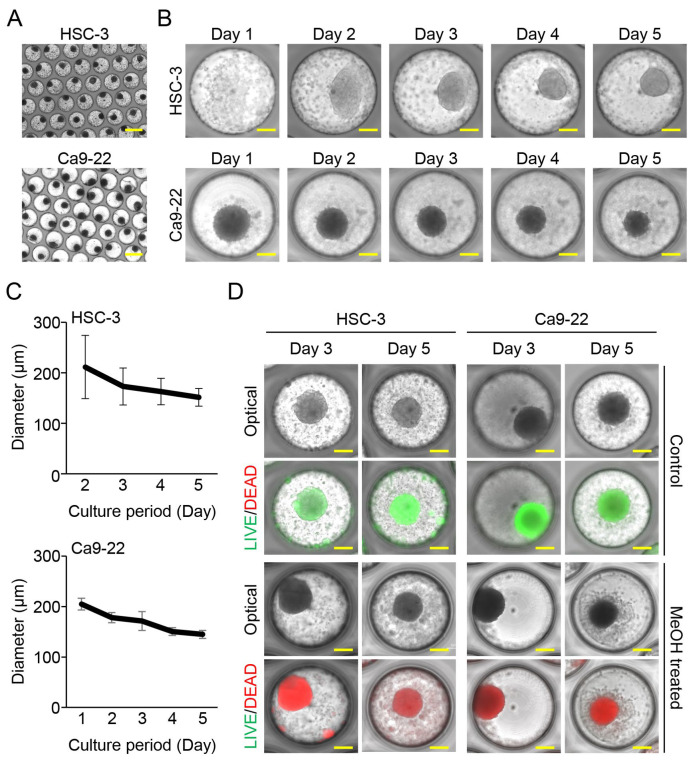
The formation of oral cancer cell spheroids in the microwell device. (**A**,**B**) Representative microscopic images with low ((**A**); scale bars: 500 μm) and high ((**B**); scale bars: 100 μm) magnifications. (**C**) The diameter of oral cancer cells spheroids (*n* = 3). (**D**) Representative microscopic images of live and dead cells of spheroid. Methanol-treated cells (MeOH treated) were used as a positive control for dead cells. Scale bars: 100 μm.

**Figure 2 cancers-15-05162-f002:**
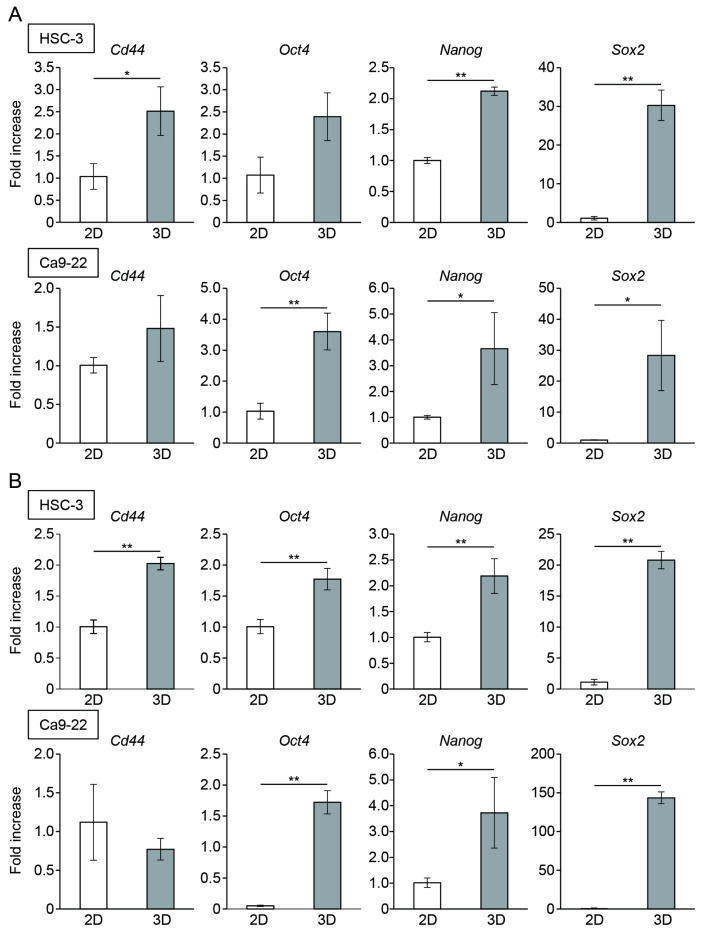
The expression of CSC marker genes in oral cancer cell spheroids. mRNA level of CSC markers in oral cancer cells cultured in the monolayer (2D) or microwell device (3D) for three (**A**) and five (**B**) days (*n* = 3). * *p* < 0.05, ** *p* < 0.01 (Student’s *t*-test).

**Figure 3 cancers-15-05162-f003:**
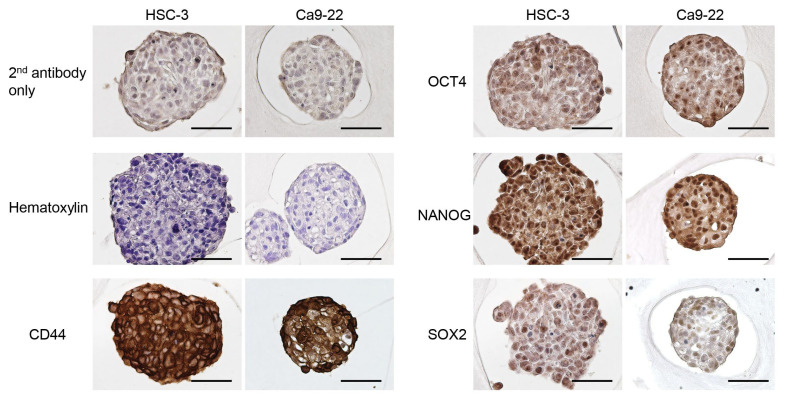
The expression and distribution of CSC marker proteins in the oral cancer cell spheroids. Representative immunohistochemical image for CSC markers in oral cancer cells cultured in the microwell device for three days. The cells were incubated without primary antibodies and used as negative controls (secondary antibody only). Counterstaining was performed using hematoxylin. Scale bars: 50 μm.

**Figure 4 cancers-15-05162-f004:**
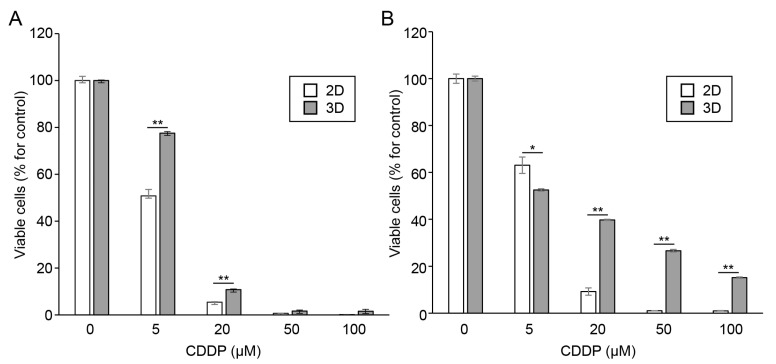
Sensitivity of oral cancer cell spheroids to anticancer drugs. The HSC-3 (**A**) and Ca9-22 (**B**) cells were cultured in monolayer (2D) or microwell chips (3D) were treated with cisplatin (CDDP; 0–100 μM) for 48 h. The proliferation of oral cancer cells was assessed using CCK-8 assay. Data are shown as the percentage of viable cells (*n* = 3). * *p* < 0.05, ** *p* < 0.01 (Student’s *t*-test).

**Figure 5 cancers-15-05162-f005:**
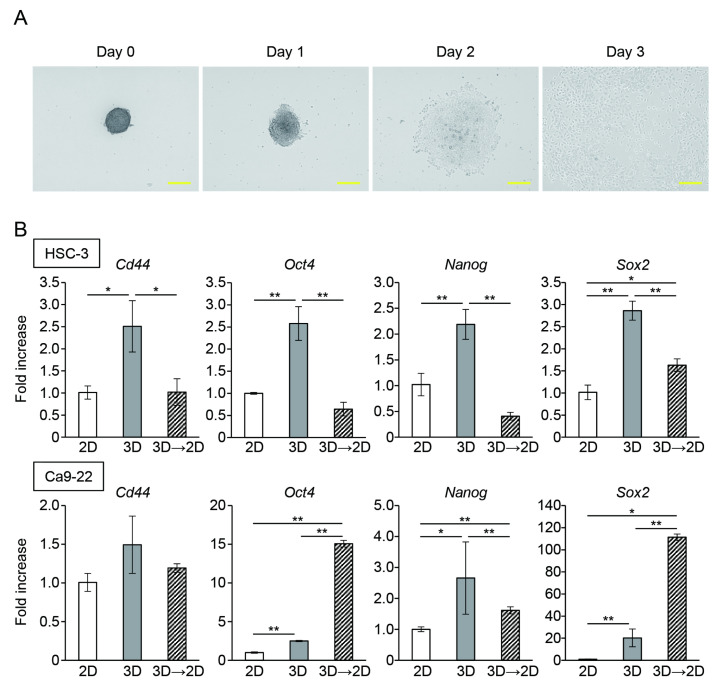
The expression of CSC marker genes in oral cancer cells outgrown from spheroids. (**A**) Representative microscopic image of outgrown cells from HSC-3 spheroids. Scale bars: 100 μm. (**B**) mRNA level of CSC marker genes in oral cancer cells cultured in monolayer (2D) or microwell device (3D) for three days or outgrown from spheroids (3D→2D) (*n* = 3). * *p* < 0.05, ** *p* < 0.01 (one-way ANOVA followed by Tukey’s test).

**Figure 6 cancers-15-05162-f006:**
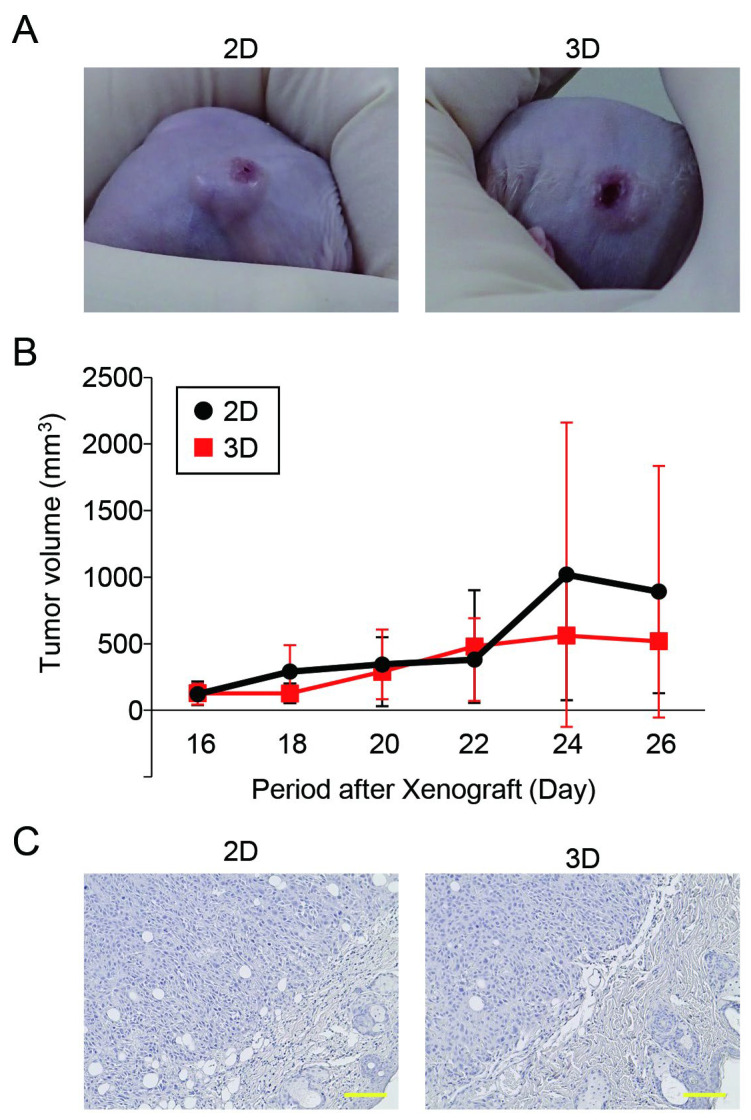
Tumor growth HSC-3-injected spheroid xenografts. (**A**) Representative photographs of back of mice injected with HSC-3 cells cultured in monolayer (2D) or microwell device (3D) on day 3. (**B**) Tumor volumes of xenograft models (*n* = 3). (**C**) Representative hematoxylin-stained images of xenograft models. Scale bars: 100 μm.

**Table 1 cancers-15-05162-t001:** Primer sequences for real-time RT-qPCR.

Gene	Primer Sequence (5’-3’)	
*β-actin*	forward	5′-GCG CGG CTA CAG CTT CA-3′
	reverse	5′-CTT AAT GTC ACG CAC GAT TTC C-3′
*Cd44*	forward	5′-TGT GCA GCA AAC AAC ACA GG-3′
	reverse	5′-TGG AGC TGA AGC ATT GAA GC-3′
*Oct4*	forward	5′-ACT CGA GCA ATT TGC CAA GC-3
	reverse	5′-TTG AAG CAA GCT GCA GAG C-3′
*Nanog*	forward	5′-GCA GAT GCA AGA ACT CTC CAA C-3′
	reverse	5′-TCG GCC AGT TGT TTT TCT GC-3′
*Sox2*	forward	5′-TGA ATG CCT TCA TGG TGT GG-3′
	reverse	5′-AGT TGT GCA TCT TGG GGT TC-3′

## Data Availability

The data presented in this article are available upon request from the corresponding author.

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
