# Peer review of "Fabrication of a Three-Dimensional Spheroid Culture System for Oral Squamous Cell Carcinomas Using a Microfabricated Device"

_cancers, 2023, doi:10.3390/cancers15215162_

Round 1

Reviewer 1 Report

In this study, the authors aimed to establish a method for producing oral cancer spheroids using the fabricated microwell device and compare the characteristics of spheroid cells with those of 2D-cultured cells. Developing a 3D culture system for oral squamous cell carcinoma using a microfabricated device has several potential applications in cancer research and treatment. The researchers found that spheroids generated using the device possessed strong cancer stem cell characteristics such as differentiation and self-renewal potential, which can help study the mechanisms of tumor pathogenesis and drug discovery under conditions similar to those of in vivo tumor microenvironments. In general, the study was well-designed and the manuscript was well-written. Minor comments:

1. The abstract should include the results of the in-vivo experiment

2. Include the number of samples in figure legends

Author Response

1. The abstract should include the results of the in-vivo experiment.

Thank you for your helpful comments. We added the results of the in-vivo xenograft experiment in the abstract of the revised manuscript (Lines 38-39).

2. Include the number of samples in figure legends.

We appreciate your important comments. We added the number of samples in figure legends of the revised manuscript(Lines 194, 209, 227, 242, and 255).

Reviewer 2 Report

The authors aimed to fabricate a polyethylene glycol-tagged microwell device that enabled spheroid formation from human oral squamous carcinoma cells, by HSC-3 and Ca9-22 cells cultured in the microwell device aggregated and generated a single spheroid per well within 24-48 h.

Their results suggest that spheroid culture system may be a high-throughput tool for producing uniform cancer stem cells.

The study covers some issues that have been overlooked in other similar topics. The structure of the manuscript appears adequate and well divided in the sections. Moreover, the study is easy to follow, but some issues should be improved. Some of the comments that would improve the overall quality of the study are:

I-) Authors must pay attention to the technical terms acronyms they used in the text

II-) Conclusion Section: This paragraph required a general revision to eliminate redundant sentences and to add some "take-home message".

Author Response

I) Authors must pay attention to the technical terms acronyms they used in the text.

Thank you for your suggestions. We carefully checked and modified the technical terms acronyms in the revised manuscript (Lines 88, 96, 189, 218, and 226).

II) Conclusion Section: This paragraph required a general revision to eliminate redundant sentences and to add some "take-home message".

We appreciate your helpful comments. We eliminated redundant sentences, add some "take-home message" and improved the Conclusion in the revised manuscript (Lines 360-364).

Reviewer 3 Report

Wataru Ariyoshi and co-workers report in this manuscript the fabrication of a 3D spheroid culture system for oral squamous cell carcinoma using a microfabricated device. This system can generate uniform oral cancer cell spheroid in large quantities. The produced spheroids showed increased expression of cancer stem cell (CSC) markers such as Cd44, Oct4, and Nanog, and resistance to anticancer drugs, indicating the outgrowth of a large population of CSCs, which would be benefit for the high-throughput studies on oral CSCs. This work was performed competently, and the results were well present. I would like to recommend publication of this manuscript in Cancers subjected to a minor revision.

Specific comments:

1.     I notice that the authors from the same group have published a few papers to report the fabrication of 3D spheroid culture systems for other types of cells, e.g., periodontal ligament cells. I understand that in the present work the authors focused on the fabrication of the 3D spheroid culture system for oral cancers cell. However, I wonder whether this system reported in this manuscript could be useful for other types of cancer cells. I suggest that the authors add a short discussion for this possibility in the manuscript.

2.     Lines 304 – 307, the authors discussed contributors to the induction of stemness of oral cancer cells in their system, however, their discussion is not convinced to me. Please add a few references there to support their discussion.

3.     Line 187, please add “, respectively” after (Ca9-22).

4.     Line 312, “enhanced” should read as “enhance”.

Author Response

1. I notice that the authors from the same group have published a few papers to report the fabrication of 3D spheroid culture systems for other types of cells, e.g., periodontal ligament cells. I understand that in the present work the authors focused on the fabrication of the 3D spheroid culture system for oral cancers cell. However, I wonder whether this system reported in this manuscript could be useful for other types of cancer cells. I suggest that the authors add a short discussion for this possibility in the manuscript.

Thank you for your comments. We mentioned the potential application of the developed device to other cancer cells and modified the discussion in the revised manuscript (Lines 269-272).

2. Lines 304 – 307, the authors discussed contributors to the induction of stemness of oral cancer cells in their system, however, their discussion is not convinced to me. Please add a few references there to support their discussion.

We appreciate your helpful suggestions. We add three references (Ref No. 38-41) and modified the Discussion in the revised manuscript (Lines 303-307).

3. Line 187, please add “, respectively” after (Ca9-22).

According to the reviewer’s comment. We added “respectively” in the revised manuscript (Line 183).

4. Line 312, “enhanced” should read as “enhance”.

We changed “enhanced” into “enhance” in the revised manuscript (Line 312).